# Assessing Strontium and Vulnerability to Strontium in Private Drinking Water Systems in Virginia

**Veronica Scott [1], Luke Juran [2,3,\*], Erin J. Ling [4] , Brian Benham [4] and Asa Spiller [4]**

1   Wisconsin State Lab of Hygiene, University of Wisconsin-Madison, Madison, WI 53705, USA; veronica.scott@wisc.edu
2   Department of Geography, Virginia Tech, Blacksburg, VA 24061, USA
3   Virginia Water Resources Research Center, Virginia Tech, Blacksburg, VA 24061, USA
4   Department of Biological Systems Engineering, Virginia Tech, Blacksburg, VA 24061, USA; ejling@vt.edu (E.J.L.); benham@vt.edu (B.B.); aspiller@vt.edu (A.S.)
\*   Correspondence: ljuran@vt.edu

**Abstract:** A total of 1.7 million Virginians rely on private drinking water (PDW) systems and 1.3 million of those people do not know their water quality. Because most Virginians who use PDW do not know the quality of that water and since strontium poses a public health risk, this study investigates sources of strontium in PDW in Virginia and identifies the areas and populations most vulnerable. Physical factors such as rock type, rock age, and fertilizer use have been linked to elevated strontium concentrations in drinking water. Social factors such as poverty, poor diet, and adolescence also increase social vulnerability to health impacts of strontium. Using water quality data from the Virginia Household Water Quality Program (VAHWQP) and statistical and spatial analyses, physical vulnerability was found to be highest in the Ridge and Valley province of Virginia where agricultural land use and geologic formations with high strontium concentrations (e.g., limestone, dolomite, sandstone, shale) are the dominant aquifer rocks. In terms of social vulnerability, households with high levels of strontium are more likely than the average VAHWQP participant to live in a food desert. This study provides information to help 1.7 million residents of Virginia, as well as populations in neighboring states, understand their risk of exposure to strontium in PDW.

**Keywords:** drinking water quality; private drinking water; strontium; Virginia; wells

## 1. Introduction

Clean drinking water is essential for a healthy life. The United Nations estimates that 502,000 people die from diarrhea due to contaminated drinking water every year [1]. In the United States, the Environmental Protection Agency (EPA) enforces the Safe Drinking Water Act (SDWA) to regulate water quality and promote public health [2]. As knowledge of contaminants increases and treatment techniques improve, the EPA adds new contaminants to the SDWA [3]. Contaminants are added to the SDWA if they are detrimental to human health, found in waters throughout the United States, and their removal or reduction poses a meaningful opportunity to improve public health [3]. The third round of new contaminant candidates included strontium due to its detrimental effects on bone growth [3,4]. However, strontium is still not regulated and, more important to this study, the SDWA only applies to *public* drinking water systems (e.g., municipal, regulated systems). Thus, a significant gap exists for the 1.7 million Virginia residents (22% of the population) and 15 million people nationally who source drinking water from private drinking water (PDW) systems such as wells, springs, and cisterns [5]. Characterizing PDW quality has the potential to benefit the health of populations reliant on these systems, especially in Virginia, where 80% of PDWs have never been tested or have been tested only

once [5]. To address these issues, this study examines PDW quality across the state of Virginia with a focus on strontium, which is relatively understudied, and on the candidate list to be added to the SDWA as a regulated contaminant.

While studies on PDW quality exist [6–15], few examine strontium at a large scale and none examine strontium in Virginia [16–18]. Data generated through the Virginia Cooperative Extension's Virginia Household Water Quality Program (VAHWQP) at Virginia Tech provide a unique opportunity to characterize, analyze, and interpret the distribution of strontium concentrations in PDW in general, and particularly in Virginia. Given this background, this study investigated the following questions:

1.　　What is the level of strontium in PDW in Virginia?
2.　　What are the physical (geologic) vulnerabilities to strontium contamination in PDW in Virginia?
3.　　What are the social (human) vulnerabilities to strontium contamination in PDW in Virginia?

*Literature Review*

Strontium is a naturally occurring alkaline earth metal element closely related to calcium [4]. The natural abundance of strontium in Earth's crust is 0.02%–0.03%, and the average concentration of strontium in freshwater globally is 0.5–1.5 mg/L [4,19]. Strontium is present in many sedimentary rocks and typically found in high levels in some calcite minerals [4,19]. Anthropogenic sources of strontium include nuclear fallout, fertilizers, and industrial manufacturing [19–23].

Health problems associated with strontium have been identified as a public health risk [3]. The EPA considers strontium nontoxic to humans under standard environmental levels [4]. However, when concentrations exceed 1.5 mg/L in water (the Health Reference Level or HRL), strontium can enter the bloodstream and replace calcium in bones, making bones brittle and eventually leading to the development of strontium rickets [4,19,24]. Strontium is particularly dangerous to children, especially infants, since their bodies have higher rates of absorption into the bloodstream while simultaneously experiencing higher rates of bone growth than adults [4,19]. Due to the potential health impacts of strontium in drinking water, the EPA added it to the Third Drinking Water Contaminant Candidate List, although a regulation decision has yet to be rendered [3]. Strontium can be removed from drinking water through treatment processes such as lime softening [25], and impacts can also be mitigated by consuming a diet rich in protein and calcium [4,19], which reduces the amount of strontium the body can absorb. Thus, populations that experience financial barriers to adopting water softening to treat drinking water or consume diets low in calcium and protein are at higher risk of strontium rickets, particularly children and infants [26].

In terms of physical vulnerabilities to strontium in PDW, Virginians are at risk of elevated levels due to the state's physical geography. There are five physiographic provinces in Virginia: Appalachian Plateau, Ridge and Valley, Blue Ridge, Piedmont, and Coastal Plain [27]. The Appalachian Plateau and Ridge and Valley are characterized by shale, slate, coal, and limestone, which constitute rock types known to be sources of strontium [19,27,28]. The Blue Ridge and Piedmont provinces are characterized by igneous and metamorphic rocks that contain strontium not readily mobilized in water due to the mineralogy [27,28]. The Coastal Plain is characterized by sandstones and unconsolidated sands, which can also contain high levels of strontium [19,27,28]. Additionally, there are >8 million acres of agricultural land use in Virginia that potentially receive phosphate fertilizers (known sources of strontium), which may leach or run off, thereby impacting surface and groundwater sources [20–23,29]. In fact, studies have shown that the type of fertilizer (e.g., monoammonium phosphate, diammonium phosphate, triple superphosphate), amount applied, and the cumulative legacy of use can contribute up to one quarter of the total strontium found in drinking water sourced from private and public wells [23].

As for social vulnerabilities, 6% of Virginians are under age five and >10% of the population lives below the federal poverty line [30,31]. Figure 1 shows childhood poverty rates, with many counties above 20% [32]. Additionally, nearly 19% of Virginians live in food deserts (areas with limited access to

nutritious food), which exceeds the national rate by more than 7% [33]. This coupling of poverty and food deserts means that many children in Virginia live below the poverty line and/or may lack access to diets high in calcium and protein that mitigate strontium damage to bones [4,19,32–34]. Another compounding social vulnerability is that 1.7 million Virginians (22% of the population) rely on PDW, which are unmonitored and unregulated for strontium, among other contaminants, by the federal and state governments [2,5].

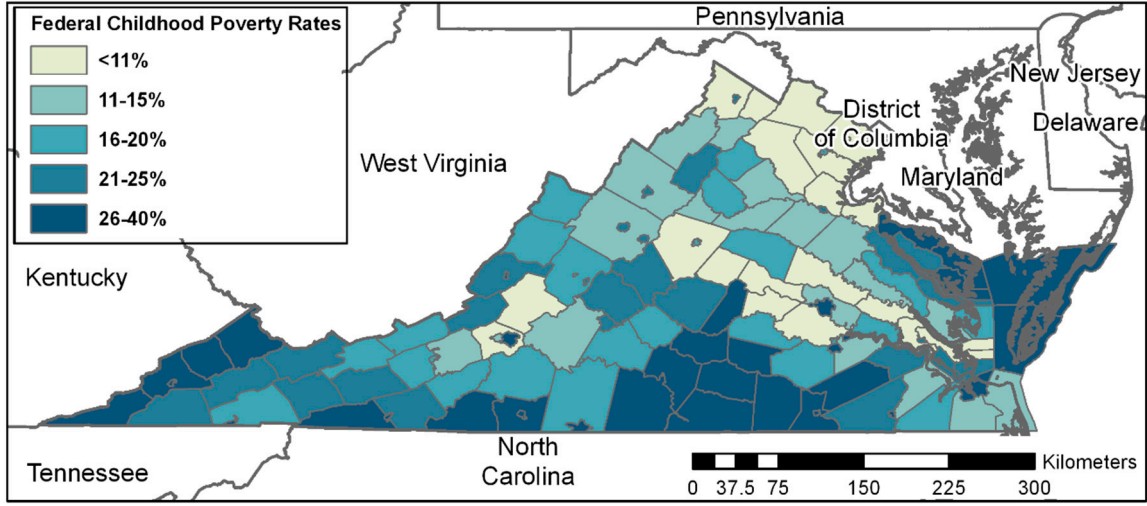

**Figure 1.** Childhood poverty rates in Virginia (% below federal poverty line).

Virginia exhibits physical and social vulnerabilities to elevated levels of strontium in drinking water, and much of the state's drinking water is supplied via unregulated PDWs. Thus, using a large, state-wide dataset, this study examines strontium in PDW in Virginia by first characterizing the concentration and spatial extent of strontium in PDW and subsequently examining physical and social vulnerabilities to strontium health risks. Thus, poverty, (lack of) access to grocery stores, number of children in an area, and a reliance on PDW interact with geology and land use to increase the risk to strontium in drinking water.

## 2. Materials and Methods

### 2.1. Data Collection

This study used water 9779 PDW samples collected in Virginia by VAHWQP between January 2014 and December 2018. VAHWQP was established as a Virginia Cooperative Extension effort in 1989 and is operated by the Department of Biological Systems Engineering at Virginia Tech [5]. The program conducts roughly 65 county-based drinking water clinics each year and has amassed data from every Virginia County [5]. Clinics provide citizens with low-cost PDW quality tests as well as objective water quality summaries and targeted PDW maintenance information [5]. Survey data reveal that 80% of samples come from PDWs that have either never been tested or have been tested only once.

Participants fill four bottles provided in a test kit with water from the tap they typically use for drinking, cooking, and other consumptive ends. One 250 mL sample is collected immediately after the tap has been turned on after at least six hours of stagnation (first draw), and a second 250 mL sample is collected after the water has run through the plumbing for at least one minute (flush draw). First and flush draw samples are collected to help understand how the water interacts with pipes, fixtures, and other components of the plumbing system and whether any contaminants (e.g., heavy metals) present are from plumbing materials or the water source itself. Two additional 125 mL samples are collected after the flush sample for analysis of remaining contaminants. Samples were transported on ice (stored between 2 and 8 °C) to Virginia Tech on the morning of collection for analysis according

to standard methods of the Environmental Protection Agency. All collection bottles were sterile, autoclaved, and arrived at the lab in <6 hours for immediate testing of total coliforms and *E. coli* using the IDEXX Colilert method. The first and flush draw samples were acidified and subsampled for metals and elemental analysis. One 125 mL sample was used to analyze pH, electroconductivity, and anions (F- and $NO_3$-N), while the other was tested for the enumeration of total coliform and *E. coli* bacteria. The geolocation, well information, and laboratory results for each sample were maintained in an Access database stored on a password protected server. Sample location addresses were converted to GPS latitude–longitude coordinates using geocod.io software and displayed at a scale preventing individual households from being identified. In order to protect the identity of participants, all identifying attributes, including names, email addresses, phone numbers, and mailing addresses, were removed from the dataset before analyses were conducted.

Supplemental data were collected from multiple sources. A geologic map shapefile containing rock types and ages was retrieved from the United States Geological Survey (USGS) [35]. The USGS National Land Cover Database 2011 raster file that identifies land cover from 2006–2011 in 30-meter resolution using 2011 Landsat imagery was also obtained [36]. Another USGS dataset used was a physiographic province shapefile digitized from Fenneman's map "Physical Divisions of the United States" that divides the country into distinct areas based on topography, rock types and structures, and geologic history [37]. Land use land cover data were downloaded as a raster dataset from the United States Department of Agriculture (USDA) [36]. TIGER line data for census tracts—which include social, economic, and demographic attributes—were gathered as a shapefile from the United States Census Bureau [38], while data on food deserts were obtained as a CSV file from USDA [39,40]. Here, food deserts are defined as census tracts with 500 residents (33% of census tract population) that live at least one mile in an urban area and 10 miles in a rural area from a grocery store [40]. Additionally, 20% or more of the population must fall below the federal poverty line or have a median family income <80% of the statewide median family income. Based on this filtering, 20.2% of Virginia census tracts are defined as food deserts [39,40].

## 2.2. Data Processing

There were 9804 PDW quality samples collected by VAHWQP between January 2014 and December 2018. However, samples lacking both a first and flush draw were removed, resulting in a total of 9779 samples. Water quality data were input into ArcMap 10.5.1 using the import x–y data tool. The TIGER line census tract shapefile and food desert supplemental CSV file were added to ArcMap and joined using the 11-digit county code number. The geographic extent of the physiographic provinces shapefile was the continental US, so the ArcMap clip tool was used to select only provinces within the boundaries of the state of Virginia using the census tract shapefile as the clip extent. Similarly, the extent of the geology shapefile was clipped to remove data unnecessary for analyses. Finally, it is important to note that this study defined 1.5 mg/L or more of strontium as a measure of "high" strontium concentration (i.e., reference level set by the EPA).

## 2.3. Statistical Analysis

Strontium data summaries were generated using RStudio 1.0.153. These data include min–max, mean, median, and first and third quartile values. Jarque Bera and Quantile–Quantile plots were used to describe the distribution of the data. Concentrations of strontium were plotted against other water quality parameters to identify relationship patterns in both first and flush draw samples. The $R^2$ value was determined for each plot using RStudio. A regression line was fit to each set of data, and $R^2$ values were used to calculate how closely data statistically fit the regression lines. These summary and exploratory analyses provided both a statistical and a visual representation of strontium in PDW samples. Kruskal–Wallis tests on water quality parameters included in the dataset were performed using RStudio. Kruskal–Wallis is a non-parametric statistical test used to compare populations with non-normal distributions. RStudio was also used to conduct comparative t-tests on samples with and

without water softeners given that they have been shown to remove strontium in addition to calcium and magnesium (i.e., hardness) [25]. These tests were performed to determine if water softeners actually remove strontium from drinking water in practice in Virginia. A 95% confidence interval was used for all statistical analyses.

*2.4. Spatial Analysis*

Anselin Local Moran's I and Global Moran's I tests were conducted using ArcMap. Anselin Local Moran's I is a technique to identify, among a geolocated dataset, statistically significant geographic locations of hot spots, cold spots, and outliers [41–43]. Global Moran's I is a measure of spatial autocorrelation based on feature locations and attributes [41–43]. The technique determines if data exhibit a statistically significant spatial distribution in terms of clustering (positive spatial autocorrelation), repelling (negative spatial autocorrelation), or randomness [41–43].

LAND_COVER data were extracted for the state of Virginia and Ridge and Valley province using the ArcMap extract-by-mask tool, and field data were stored and analyzed using Microsoft Excel 2016. Land cover data for VAHWQP samples (all samples and high strontium samples) were collected using ArcMap extract-to-point and summarize tools on the LAND_COVER field with the FID first value. The four land cover datasets, the state of Virginia, Ridge and Valley province, all VAHWQP samples, and high strontium samples were combined in Excel, and a percentage was calculated for each land cover type as well as land cover type adjacent to high strontium sample locations. To determine geologic influence on strontium contamination, a similar method was used with the exception that values were extracted using the select-by-location and summarize tools in ArcMap to select data on both UNIT_AGE and ROCKTYPE1 fields by summing the AREA field.

The food desert CSV contained several measures of poverty, three of which were used in this analysis: percent of population in poverty, percent of households without access to a vehicle (HUNV), and percent of households receiving Supplemental Nutrition Assistance Program (SNAP) benefits. SNAP is a federal program that combats food insecurity for low income individuals [40] HUNV was included to identify households that earn too much to fall below the poverty line yet lack access to a vehicle. Individuals in these households may have difficulty accessing diets high in calcium and protein and may be less likely to be able to afford water treatment mechanisms if high strontium concentrations are present in their PDW. Percent of population <19 (i.e., vulnerable age cohort) in each census tract was calculated using the calculate field tool in ArcMap as a rate of the TractKids field and POP2010 field. Percent of HUNV and households receiving SNAP benefits were also calculated using the calculate field tool using the TractHUNV (for households without access to a vehicle), TractSNAP (for households collecting SNAP benefits), and OHU2010 (for number of households in each tract) fields.

Percent of population <19, percent in poverty, percent of HUNV, percent living in a food desert, and percent receiving SNAP benefits were displayed in ArcMap using five color categories with approximately the same number of counties in each color bracket while still maintaining whole numbers as percentage breaks. The Anselin Local Moran's I tool was run on all five categories to identify statistically significant geographic distributions of "hotspots" (high-high clusters), "cold spots" (low-low clusters), and outliers (high-low, low-high, and high values surrounded by low values and vice versa). The select-by-attribute tool was used in ArcMap to select high-high clusters and high-low outliers for each of the categories for the state of Virginia. The total number of census tracts and number of selected census tracts for each category were input to Excel. The select-by-location tool was then used to categorize census tracts based on location in the Ridge and Valley province, participation in VAHWQP, and presence of high strontium values. For each of these three selections, the select-by-attribute tool was used to determine the number of high-low outliers and high-high clusters for each subset of census tracts. Finally, the total number of census tracts and number of selected tracts for each category, and each subset of tracts were input to Excel where percentages of census tracts were calculated.

## 3. Results

### 3.1. Level of Strontium in PDW systems in Virginia

Of 9779 PDW samples, 122 first and 124 flush draw samples exceeded 1.5 mg/L—a high level that exceeds the HRL set by the EPA (Table 1). The highest first draw concentration was 28.75 mg/L, and the highest flush draw concentration was 29.71 mg/L. However, the vast majority (close to 99%) of samples had strontium concentrations below the reference level. Jarque Bera, Quantile–Quantile analyses, and box and whisker plots determined that the distribution of strontium concentrations in both first and flush draw samples were non-normal with a left skew, indicating that outliers existed. The mean was nearly four times larger than the median, since mean was more sensitive to outliers. A correlation of strontium concentrations in the first and flush draw samples resulted in an $R^2$ of 0.984, indicating a high degree of correlation between the two (i.e., first and flush samples were similar in terms of strontium concentrations). Similarly, Kruskal–Wallis tests produced a *p*-value of <0.001, further indicating that first and flush samples were statistically similar. Given these results, remaining analyses were conducted only on first draw samples.

**Table 1.** Statistical summary of strontium first and flush draw samples.

|  | First Draw (mg/L) | Flush Draw (mg/L) |
| --- | --- | --- |
| Range | 0–28.750 | 0–29.710 |
| Mean | 0.158 | 0.160 |
| Median | 0.043 | 0.044 |
| 1st Quartile | 0.014 | 0.014 |
| 3rd Quartile | 0.109 | 0.109 |
| Standard Deviation | 0.808 | 0.833 |
| High Strontium Samples (≥1.5 mg/L) | 122 | 124 |
| Total Samples | 9779 | 9779 |

A total of 2350 samples were from PDW with water softeners. Of the 2350 samples treated with water softeners, 30 (1.28%) contained high strontium. Of the 7249 samples without water softener treatment, 92 (1.27%) contained high strontium (note that 200 samples failed to report whether a softener was or was not used). Samples with water softeners had an average concentration of 0.096 mg/L, while samples without softeners had an average concentration of 0.190 mg/L, double the strontium concentration of PDW with softeners. Comparative t-tests on strontium concentrations identified statistically significant differences between PDW with and without softeners (p < 0.001), with samples with water softeners exhibiting significantly less concentrations of strontium.

Locations of the 122 first draw samples with high strontium concentrations (≥1.5 mg/L) were plotted (Figure 2). Seventy percent of high concentrations were found in the Ridge and Valley province that parallels Virginia's western/northwestern border (Figure 2). A Global Moran's I test for clustering produced a *p*-value of <0.001 and z-score of 24.574, indicating a <1% chance that spatial distributions (i.e., clusters) are a result of randomness. A Local Moran's I test mapped the clusters (Figure 3), demonstrating that geographic locations with high concentrations of strontium are near other high concentration samples; the pattern of high-high and high-low clusters mirrors the southwest–northwest trend of high strontium displayed in Figure 2.

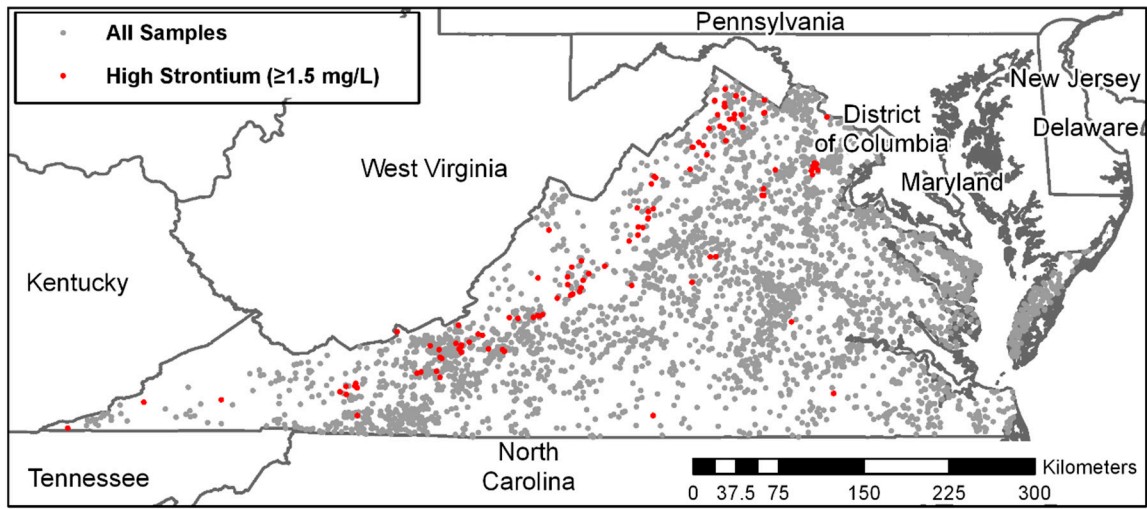

**Figure 2.** Map of 9779 Virginia Household Water Quality Program (VAHWQP) private drinking water (PDW) samples collected 2014–2018.

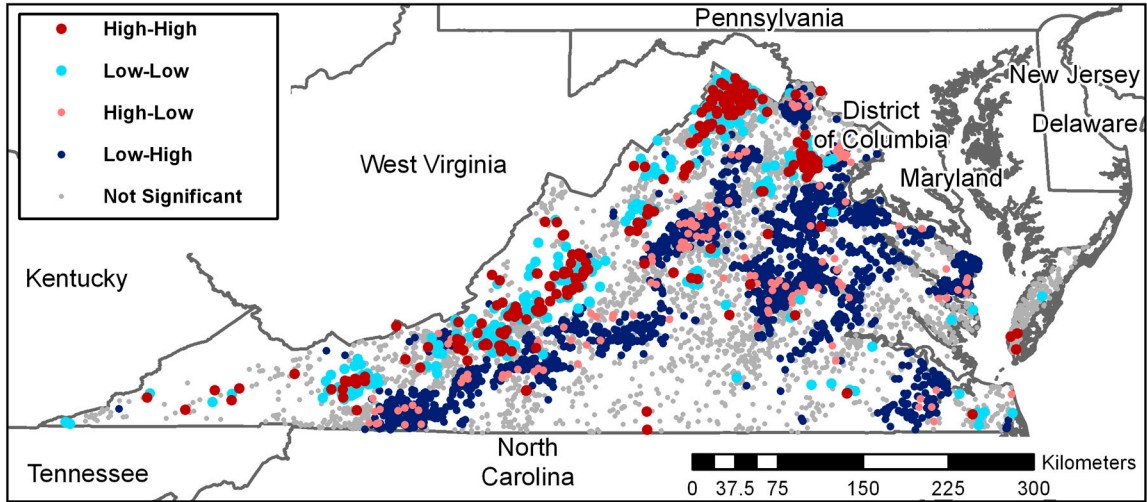

**Figure 3.** Local Moran's I cluster analysis of strontium concentrations in first draw samples.

### 3.2. Physical Vulnerabilities to Strontium Contamination

Strontium was weakly correlated ($R^2 > 0.100$, $p < 0.500$) with calcium and magnesium but had no correlation with 29 other water quality parameters ($R^2 < 0.100$, $p > 0.500$) that were tested. Jarque Bera tests and Quantile–Quantile plots were used to determine the normality of sample concentrations for the 29 water quality parameters, and Kruskal–Wallis tests determined the statistical correlation between strontium and each analyzed water quality parameter. Results indicated that 16 water quality parameters were statistically similar to strontium, while 16 parameters were also positively correlated to a statistically significant degree. However, only three parameters satisfied both of these requirements (Table 2).

Land use land cover analysis found that the highest number of high strontium samples were from deciduous forests, hay/pasture, and developed open spaces. However, medium intensity developed land, herbaceous, developed open space, and low intensity developed land had more high strontium samples on a percentage basis (Table 3). While herbaceous and medium intensity developed land cover types had a high percentage of pixels with high strontium samples, the total number of high strontium samples was too low for a robust analysis and was not considered further. The Ridge and Valley province, where most of the high strontium samples occurred, had higher rates of deciduous forest and hay/pasture land use land cover compared to the rest of the state.

**Table 2.** Statistical analysis of water quality parameter relationships to strontium.

| Parameter | First Draw | | | | Flush Draw | | | |
|---|---|---|---|---|---|---|---|---|
| | $R^2$ | r | Jarque Bera $p$-Value | Kruskal Wallis $p$-Value | $R^2$ | r | Jarque Bera $p$-Value | Kruskal Wallis $p$-Value |
| Mg | 0.092 | 0.717[*] | <0.000 | 0.000[**] | 0.094 | 0.722[*] | <0.000 | 0.000[**] |
| Ca | 0.122[*] | 0.813[*] | <0.000 | 0.000[**] | 0.113[*] | 0.803[*] | <0.000 | 0.000[**] |
| Hardness | 0.125[*] | 0.116 | <0.000 | 0.250 | 0.119[*] | 0.797[*] | <0.000 | 0.094[**] |
| Sr | 0.984[*] | 0.971[*] | <0.000 | 0.000[**] | N/A | N/A | <0.000 | N/A |

[*] Significant correlation $R^2 > 0.100$ or $r > 0.700$. [**] Significant $p$-value <0.05. The following parameters were found not to have a significant relationship to strontium: Al, Ag, As, Ba, Cd, Cl, Co, Cr, F, Fe, K, Mn, Mo, Na, Ni, P, Pb, pH, Se, Si, SO$_4$, Sn, Ti, U, electroconductivity, and presence of *E. coli*.

**Table 3.** Results of land use land cover (LULC) analysis.

| Land Use Land Cover | % LULC Occurring in Ridge and Valley | % LULC Occurring on High Sr Samples (≥1.5 mg/L) |
|---|---|---|
| Cultivated Crops | 6.15 | 0.91 |
| Evergreen Forest | 11.82 | 1.10 |
| Developed Medium Intensity | 21.38 | 2.08 |
| Herbaceous | 22.37 | 2.06 |
| Developed Open Space | 22.97 | 1.19 |
| Developed Low Intensity | 25.91 | 1.84 |
| Deciduous Forest | 35.51 | 1.09 |
| Hay/Pasture | 37.25 | 1.71 |
| Total | 26.91 | 1.25 |

Virginia is underlain by geologic formations spanning more than 500 million years from the Cambrian to the Quaternary. Sedimentary rocks such as shale, sand, sandstone, and gravel are the most common rock types in Virginia. The rock types most common where VAHWQP samples were collected are sand, gravel, and shale, which is similar to the common rock cover in the state as a whole (Table 4). The greatest number of high strontium samples were collected from locations where Cambrian and Ordovician age rocks are found (35 of 102 high strontium samples). The Ridge and Valley province, where most of the high strontium samples occurred, has >90% of the shale, dolomite, limestone, and black shale in the state. High strontium samples were most commonly found on limestone, dolomite, sandstone, shale, and black shale. Alluvium, anorthosite, conglomerate, quartz monzonite, metasedimentary, and mylonite rock types each had only one high strontium sample and were thus excluded from analysis. High strontium samples occurred on similar rock types and with a similar spatial distribution as the most common rock types in the Ridge and Valley province.

**Table 4.** Results of rock formation analysis.

| Rock Age | # of Formations in Virginia | # of Formations in Ridge and Valley | % of Formation Area in Ridge and Valley | # of Formations Under VAHWQP Samples | # of Formations Under High^ Sr VAHWQP Samples | % of Formation Area Under High^ Sr VAHWQP Samples |
|---|---|---|---|---|---|---|
| Cambrian | 591 | 246 | 49.3 | 162 | 14 | 21.3 |
| Cambrian–Ord | 148 | 108 | 84.4 | 46 | 8 | 26.1 |
| Ordovician | 442 | 374 | 91.1 | 100 | 13 | 42.5 |
| Ordovician–De | 10 | 10 | 100 | 2 | 1 | 27.8 |
| **Rock Type** | | | | | | |
| Black Shale | 98 | 97 | 99.9 | 17 | 2 | 65.1 |
| Dolomite | 234 | 223 | 99.3 | 82 | 15 | 32.7 |
| Limestone | 223 | 216 | 99.9 | 50 | 13 | 52.2 |
| Sand | 695 | 0 | 0 | 171 | 0 | 0 |
| Sandstone | 557 | 268 | 65.6 | 72 | 9 | 14.0 |
| Shale | 468 | 394 | 94.5 | 101 | 12 | 28.5 |

^ High strontium defined as ≥1.5 mg/L.

### 3.3. Social Vulnerabilities to Strontium Contamination

Fifteen percent of Virginia census tracts (296 of 1971 tracts) were determined to be food deserts (Figure 4), and 20.2% of census tracts with high strontium samples are located in food deserts (Table 5). Analyses indicate that census tracts with VAHWQP samples have lower rates of households without access to a vehicle and a lower percentage of population below age 19 (Table 5). Additionally, census tracts that participated in VAHWQP have lower rates of households receiving SNAP benefits while census tracts with high strontium concentrations have even lower rates of households receiving SNAP benefits. Census tracts with high levels of strontium exhibited lower rates of households in poverty when compared to the remaining tracts participating in VAHWQP, tracts in the Ridge and Valley province, and the state as a whole. Spatial patterns in Figure 5 reveal that the northern portion of Virginia is generally wealthier with more children, higher rates of households with vehicles, and lower rates of households on SNAP benefits compared to the rest of Virginia. Each measure of vulnerability indicates slightly different regions as most vulnerable to the health effects of strontium, but in general the far southwest of Virginia along with Greensville, Sussex, and Southampton counties, which are clustered near the southern border of Virginia south of the city of Richmond, were identified as the most socially vulnerable areas.

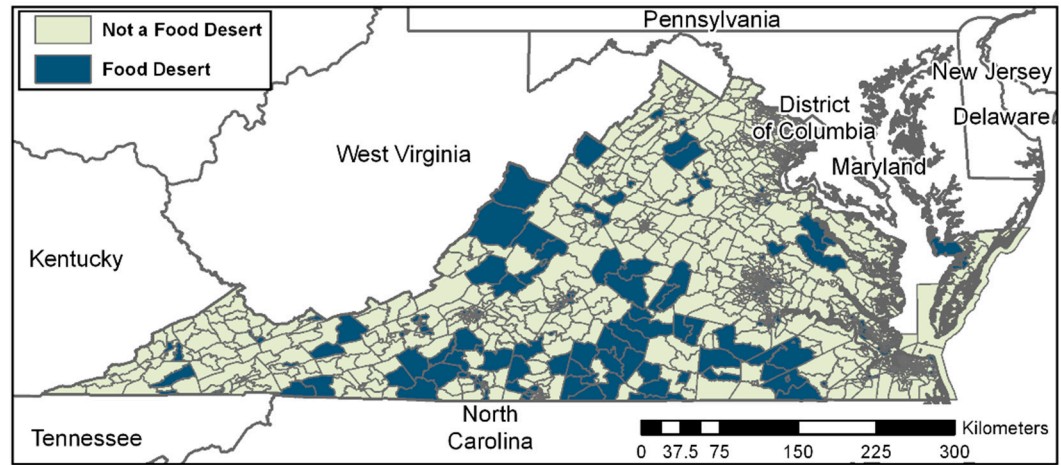

**Figure 4.** Food deserts by census tract.

**Table 5.** Summary of social vulnerability analysis.

| | % Census Tracts in VA | % Tracts in Ridge and Valley | % Tracts Participating in VAHWQP | % Tracts with High Sr Samples (≥1.5 mg/L) |
|---|---|---|---|---|
| High[*] Poverty Rate | 23.2 | 21.8 | 13.0 | 11.6 |
| High[*] Rate of Households on SNAP[^] | 23.2 | 20.5 | 13.2 | 2.9 |
| High[*] Rate of Households w/o Vehicle | 16.8 | 5.9 | 3.8 | 0.0 |
| High[*] % of Pop. <19 | 26.9 | 5.9 | 23.4 | 11.6 |
| Classified as Food Desert | 15.0 | 19.6 | 14.6 | 20.3 |

[*] Local Moran's I test identified "high" as a statistically significant high–high cluster or high–low outlier census tract. [^] Supplemental Nutrition Assistance Program.

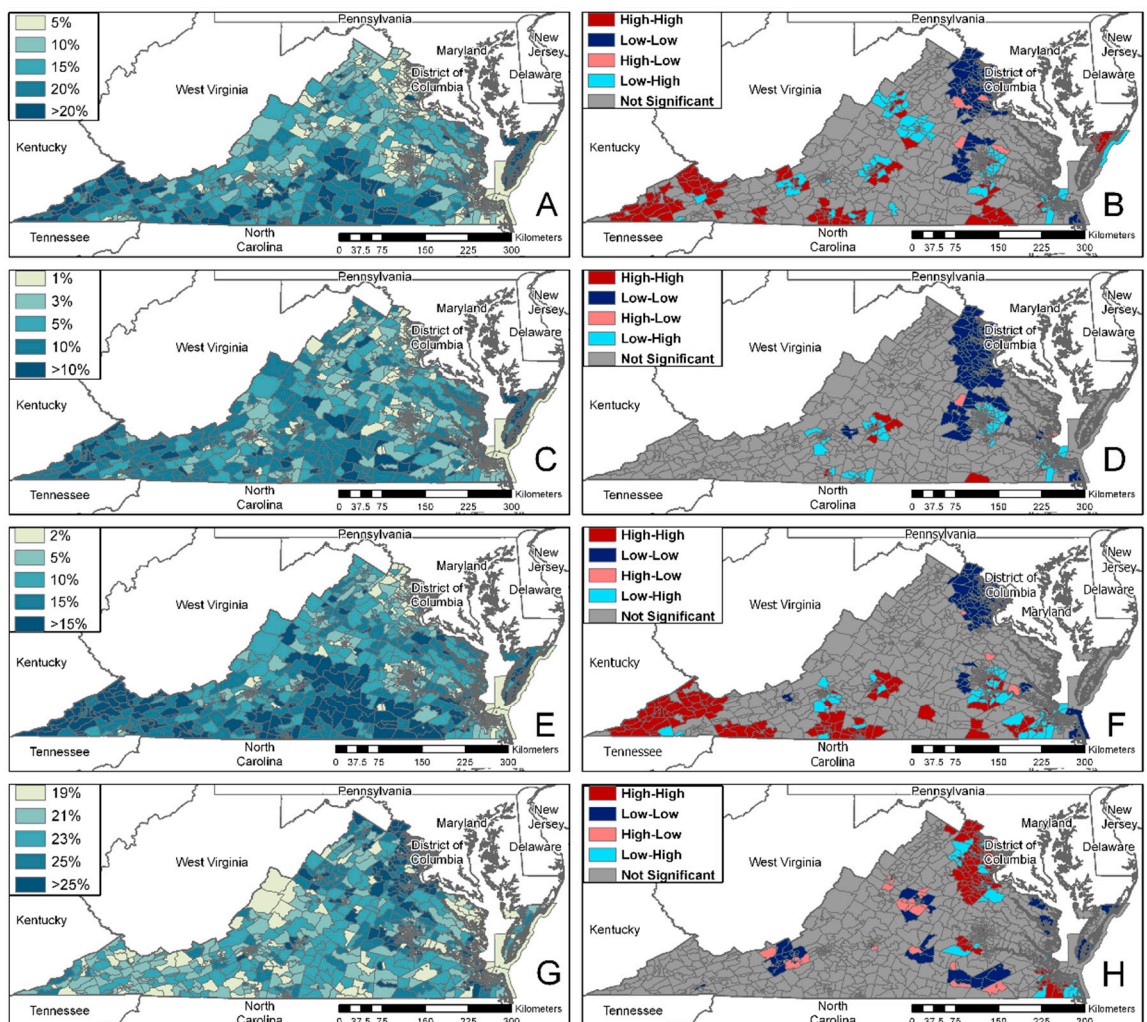

**Figure 5.** Poverty related risk factors by census tract: (**A**) Distribution of poverty rates; (**B**) Cluster analysis of poverty rates; (**C**) Distribution of rate of households without access to a vehicle; (**D**) Cluster analysis of rate of households without access to a vehicle; (**E**) Distribution of rate of households receiving SNAP benefits; (**F**) Cluster analysis of rate of households receiving SNAP benefits; (**G**) Percent of population below age 19; and (**H**) Cluster analysis of percent of population below age 19.

## 4. Discussion

Only 1.2% of PDW samples exhibited high concentrations of strontium, defined here as at or above the EPA health reference level of 1.5 mg/L. While this rate suggests that most Virginians are not at risk from strontium contamination in PDW, the statistically significant pattern of high clusters and high outliers indicates that populations in the Ridge and Valley province (e.g., Frederick, Shenandoah, Rockbridge, and Botetourt counties) that rely on PDWs have a higher probability (3.5%) of exceeding the HRL. There is also a cluster of high strontium concentrations around Prince William County in Northeastern Virginia. After further examination, samples in Prince William County had high rates of calcium, magnesium, and hardness, which is similar to what was observed in the Ridge and Valley. There were 2350 samples from households that reported using water softeners. Of these samples, 1.3% exceeded the HRL for strontium, which mirrors the rate of the entire group. Here, it is critical to note that nearly one quarter of households are using water softeners that have been shown to reduce the amount of strontium in treated water; however, in this case softeners are reducing the amount of strontium but not removing enough to drop levels below the reference level.

In terms of land use land cover, >80% of sample locations with high strontium overlay with hay/pasture, developed open space, and developed low intensity types, suggesting that human development and human–environment interactions may impact strontium concentrations in PDWs. Interestingly, high intensity development land use had no high strontium samples. We argue that this is because most homes in that land use type are served by public/municipal water systems and therefore do not have PDWs. Agriculture (cultivated crops and hay/pasture)—another land use defined by human intervention—appeared to have a large impact on strontium given that 42.5% of all high strontium samples overlay on land designated as agricultural. This indicates that phosphate fertilizer applications on agricultural land may impact strontium concentrations in PDWs. Phosphate fertilizers—known sources of strontium—can leach and enter surface and groundwater sources [20–23], and studies have demonstrated that type of fertilizer (e.g., monoammonium phosphate, diammonium phosphate, triple superphosphate), amount applied, and legacy of use can contribute up to 25% of the total strontium in drinking water sourced from private and public wells [23]. Ultimately, there is a high rate of high strontium occurrence in areas associated with human intervention (>80%), and more research should be conducted to determine if this risk is connected to fertilizer applications not only for agriculture, but also in urban spaces for purposes of lawn care and to increase the aesthetics of parks, golf courses, and other landscapes.

The five most common rock types in Virginia (limestone, dolomite, sandstone, shale, and black shale) account for 90% of high strontium concentration samples. The southwest–northeast spatial pattern of these rock types is similar to the southwest–northeast spatial pattern of high strontium samples, and these patterns mirror and are particularly evident in the Ridge and Valley province. This indicates that geology influences strontium concentrations. Therefore, this study finds an association between rock type and strontium concentrations but cannot determine with certainty if rock types are the source of strontium without samples of aquifer source rock for each PDW. However, this does indicate high physiographic (geologic) vulnerability for households with PDWs located on those particular rock types.

The only factor that revealed social vulnerability in areas of high strontium concentration was food deserts. This is significant because access to a nutritious diet high in calcium and protein is essential to counteract adverse health impacts from ingesting excess strontium, and because Virginia has a higher rate of people living in food deserts compared to the national average [31]. While physical vulnerability factors follow the Ridge and Valley province, social vulnerability factors do not overlap the physically vulnerable regions. Findings indicate that the southwest corner of Virginia (Appalachian Plateau) and southside/southeast Virginia (e.g., Greensville, Sussex, and Southampton counties) have unusually high rates of poverty (defined by federal poverty rates), percent of households receiving SNAP benefits, and percent of households without access to a vehicle. Census tracts in northern Virginia have the highest percentage of children population (below age 19) but are also relatively wealthy. Less than 6% of census tracts in the Ridge and Valley province have a high percentage of children population compared to 23% for all VAHWQP sample locations and 11% for high strontium sample census tracts.

In terms of addressing issues of social vulnerability, this study has the most potential to benefit census tracts with higher percentages of children. Interestingly, census tracts with higher percentages of children actually participated in VAHWQP at greater rates; however, these tracts also happen to be relatively wealthier. One reason for this economic bias is that VAHWQP is a voluntary program with a monetary cost of approximately $60 for water quality testing, which can present an economic barrier to participation among relatively less wealthy households. Furthermore, samples must be collected according to specific instructions and dropped at specific locations on specific dates at specific times—which presents additional barriers related to access to transportation, mobility, economics, education, class, and social location broadly. Another caveat is that most samples are from participants who own their home and thus own the PDW—this makes sense as such participants are likely in charge of PDW operation and maintenance and thus may have greater incentive to participate. This caveat may bias the dataset in favor of relatively wealthier households and inadvertently skew the dataset

away from renters and residential contexts such as mobile home parks [5]. These populations may be experiencing issues with their PDW but lack the (economic and political) capacity, agency, or perhaps feel apprehensive (out of fear of retaliation) to seek water quality testing through VAHWQP [44,45]. Given this background, locations where many VAHWQP samples were collected tend to have lower poverty rates, lower SNAP benefit rates, and greater rates of access to a vehicle compared to the Virginia as a while. On the other hand, since poverty and percent of children population may be biased by voluntary, economic, and other social barriers to VAHWQP entry, this study therefore likely underreports social vulnerability in locations with high poverty and/or rental rates. Other factors that may lead to underreporting social vulnerability are representative samples from populations who lack trust in the government, scientists, and institutions such as VAHWQP and those who perceive their PDW as "safe" to begin with [45,46].

## 5. Conclusions

With 1.7 million Virginians relying on PDW and few knowing the status of their water quality, there is urgent need to understand the public health risks these residents face. Strontium has the potential to cause damage to health (especially children) and as such is being considered for regulation by the EPA with a reference level of 1.5 mg/L. Strontium can be removed via water softening, so it is critical to inform the public of the risks, so they can mitigate health hazards and keep their families and communities safe. Thus, this study provides an initial characterization of the prevalence, concentration, spatial distribution, and physical and social vulnerabilities associated with strontium in PDW in Virginia in order to inform the public and address an understudied issue of public health concern. While studies on PDWs exist [6–15], few examine strontium at a large scale and none examine strontium in Virginia [16–18]; further, 80% of the PDWs were either never tested or have been tested only once.

Overall, areas of Virginia with high concentrations of strontium generally exhibit low social vulnerability to strontium, with the exception of those living in a food desert. This is particularly important because diets high in calcium and protein counteract the effects of drinking water with high levels of strontium. Individuals living in the Ridge and Valley province have the highest physical (geologic) vulnerability to elevated strontium in PDW, and prevailing rock types (limestone, dolomite, sandstone, shale, and black shale) in the region also tend to contain high levels strontium. Based on findings, this study postulates that there is greater physical vulnerability to strontium in PDW when those rock types are present. Agricultural and developed land use were also linked to high concentrations of strontium, indicating increased physical vulnerability to strontium in PDW when human interventions/activities are present on the landscape. However, further work should be conducted to specifically examine the origin of strontium in PDW and the extent human interventions impact strontium in PDW.

This research can assist Virginians with PDW to understand how they may be vulnerable to the health impacts of strontium and, subsequently, how to take actions to protect their families and communities from the risks. Broader impacts of this research, however, reach far beyond those who rely on PDW in Virginia. All residents with PDW in the United States—especially those in states near Virginia that contain part of the Ridge and Valley province (e.g., Alabama, Georgia, Tennessee, West Virginia, Maryland, Pennsylvania, New York)—could benefit from this research. Furthermore, since a portion of public drinking water provided by local water authorities in the Ridge and Valley province is sourced from groundwater, many Virginia residents (and residents in similar states with geophysical landscapes) are consuming the same source water as PDW consumers. Thus, consumers of water from PDW *and* public supplies in the Ridge and Valley province and adjacent states are both physically vulnerable to strontium contamination. This again underscores the need for strontium to be included in the EPA SDWA and additional outreach and testing for water consumers that rely on PDWs. In particular, consumers that should be targeted with clear information on the health risks of drinking unsafe water are those who cannot afford testing, renters, and those with perceptions that their water is "safe" or who lack trust in science and institutions. This is critical as such populations

may be unwittingly exposed to contaminants. Communication of science and health risks associated with drinking water is a field ripe for advancement—especially given recent international events such as Flint, Michigan. Thus, we argue for community-level approaches demonstrated by Dettori et al. [46], Arcipowski et al. [47], and Virginia Cooperative Extension's VAHQWP in their county level PDW clinics and informational material (see https://www.wellwater.bse.vt.edu/vahwqp.php). These approaches of building a "trust ecology," combined with accessible resources for unbiased water quality testing (e.g., the tests conducted through VAHWQP), represent potential steps in the right direction.

**Author Contributions:** Conceptualization, V.S., L.J., and E.J.L.; Data curation, V.S., L.J., E.J.L., B.B., and A.S.; Formal analysis, V.S., L.J., E.J.L., and A.S.; Funding acquisition, L.J., E.J.L., and B.B.; Investigation, V.S., L.J., E.J.L., and B.B.; Methodology, V.S., L.J., and E.J.L.; Project administration, L.J., E.J.L., B.B., and A.S.; Resources, L.J., E.J.L., B.B., and A.S.; Software, V.S., L.J., and A.S.; Supervision, L.J., E.J.L., and B.B.; Validation, V.S., L.J., E.J.L., B.B., and A.S.; Visualization, V.S. and L.J.; Writing—original draft, V.S., L.J., E.J.L., and B.B.; Writing—review and editing, V.S., L.J., E.J.L., B.B., and A.S.. All authors have read and agreed to the published version of the manuscript.

**Funding:** This research received no external funding.

**Acknowledgments:** The authors would like to acknowledge Kelly Peeler of the Biological Science Engineering Water Quality Laboratory, Jeff Parks of the Department of Civil and Environmental Engineering, Yang Shao of the Department of Geography, and Madeline Schreiber of the Department of Geosciences at Virginia Tech for their contributions to this research.

**Conflicts of Interest:** The authors declare no conflict of interest.

## Abbreviations

| | |
|---|---|
| (EPA) | Environmental Protection Agency |
| (HRL) | Health Reference Level |
| (PDW) | Private Drinking Water |
| (SDWA) | Safe Drinking Water Act |
| (SNAP) | Supplemental Nutrition Assistance Program |
| (USDA) | United States Department of Agriculture |
| (VAHWQP) | Virginia Household Water Quality Program |

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
