# Peer review of "Assessing Strontium and Vulnerability to Strontium in Private Drinking Water Systems in Virginia"

_water, doi:10.3390/w12041053_

Round 1

Reviewer 1 Report

 I had a few comments. Please see attached file. My comments are in red.

Author Response

Reviewer 1:

Line 57: rocks and typically found in high levels in some calcite minerals [4, 19].

This has been corrected.

Line 73: In terms of somatic vulnerabilities to strontium in PDW, Virginians are at risk of elevated levels

We have left this text unchanged (‘physical’), which we believe is clearer to the reader and more in line with the variables of geology, rock type, etc.

Line 99: Virginia exhibits somatic and social vulnerabilities to elevated levels of strontium in drinking

We have left this text unchanged (‘physical’), which we believe is clearer to the reader and more in line with the variables of geology, rock type, etc.

Line 102: concentration and spatial extent of strontium in PDW and subsequently examining somatic and….

We have left this text unchanged (‘physical’), which we believe is clearer to the reader and more in line with the variables of geology, rock type, etc.

Line 124: analysis. One 125 mL sample is used to analyze pH, electroconductivity, and anions (Fl and NO₃-N), …….. I am not sure if the authors are referring to Flerovium (FI) or Fluoride (F-). This needs to be revisited.

This should be Fluoride. We have corrected here as well as in the Table 2 notes.

Line 334-335: l. This indicates that phosphate 335 fertilizer applications on agricultural land may impact strontium concentrations in PDWs. It would be very insightful and important if the authors could in a very succinct way indicate by which process phosphate could play a role.

This was explained earlier in the manuscript, but we have now mentioned it here as well. The following sentence (which includes several references) has been added: “Phosphate fertilizers—known sources of strontium—can leach and enter surface and groundwater sources [21-24], and studies have demonstrated that type of fertilizer (e.g., monoammonium phosphate, diammonium phosphate, triple superphosphate), amount applied, and legacy of use can contribute up to 25% of the total strontium in drinking water sourced from private and public wells [22, 29].”

Reviewer 2 Report

The paper presents some relevance for the subject.

Still, reviews should be performed to increase the scientific degree of the paper.

In this regard, in the following, I will leave my reviews:

  1. A schematic figure about the methodology used is missing.
  2. The conclusions are too weak. How they are now I cannot see how it contributes to the advance of the thematic field. In this regard, other researches and studies should be exposed, confronted/compared to create a discussion in order to enrich the state-of-the-art and consequently further develop the field.

Author Response

Reviewer 2:

The paper presents some relevance for the subject. Still, reviews should be performed to increase the scientific degree of the paper. In this regard, in the following, I will leave my reviews:

  1. A schematic figure about the methodology used is missing.

A schematic is not required by the Journal. We have detailed the methodology step-by-step and have numerous figures and tables that align with the methods to make everything clear. We will defer to the Guest Editor/Journal as to whether they feel a schematic should be included.

  1. The conclusions are too weak. How they are now I cannot see how it contributes to the advance of the thematic field. In this regard, other researches and studies should be exposed, confronted/compared to create a discussion in order to enrich the state-of-the-art and consequently further develop the field.

Thank you for your comment. First, the following has been added to the Discussion—which is then expounded upon in the Conclusion—to add context and nuance: “Other factors that may lead to underreporting social vulnerability are representative samples from populations who lack trust in the government, scientists, and institutions such as VAHWQP and those who perceive their PDW as ‘safe’ to begin with [46-47].” Note that this addition includes additional references to the literature as suggested. Next, the following has been added to the Conclusion to strengthen the novelty / state-of-the-art nature of the study: “While studies on PDWs exist [6-15], few examine strontium at a large scale and none examine strontium in Virginia [16-18]; further, 80% of the PDWs were either never tested or have been tested only once.” This statement exposes other studies in that no such similar study has been conducted to date. Finally, the following has also been added to strengthen the Conclusion: “In particular, consumers that should be targeted with clear information on the health risks of drinking unsafe water are those who cannot afford testing, renters, and those with perceptions that their water is ‘safe’ or who lack trust in science and institutions. This is critical as such populations may be unwittingly exposed to contaminants. Communication of science and health risks associated with drinking water is a field ripe for advancement—especially given recent international events such as Flint, Michigan. Thus, we argue for community-level approaches demonstrated by Dettori et al. [46], Arcipowski et al. [47], and Virginia Cooperative Extension’s VAHWQP in their county level PDW clinics and informational material (see https://www.wellwater.bse.vt.edu/vahwqp.php). These approaches of building a ‘trust ecology,’ combined with accessible resources for unbiased water quality testing (e.g., the tests conducted through VAHWQP), represent potential steps in the right direction.” This addition offers nuance, recommendations, and a way forward while also engaging the literature.

Reviewer 3 Report

This manuscript is very well written and provides very important findings and helpful information about private water quality data of national and international significance, and implications on public health. Below are a few comments and suggestions for the authors' considerations.

Line 125: Provide more details about coliform/E. coli sample collection, including test bottles sterility, transport, holding and analysis times.

Line 334: Explain reasoning for focusing only on phosphate fertilizers, and not referring to other types of fertilizers, such as nitrate fertilizers that could be applied.

Author Response

Reviewer 3:

This manuscript is very well written and provides very important findings and helpful information about private water quality data of national and international significance, and implications on public health. Below are a few comments and suggestions for the authors' considerations.

Line 125: Provide more details about coliform/E. coli sample collection, including test bottles sterility, transport, holding and analysis times.

We have expanded the description of sampling collection and analysis by adding the following supplementary text: “Samples were transported on ice (stored between 2-8°C) to Virginia Tech on the morning of collection for analysis according to Standard Methods of the Environmental Protection Agency. All collection bottles were sterile, autoclaved, and arrived to the lab in <6 hours for immediate testing of total coliforms and E. coli using the IDEXX Colilert method.”

Line 334: Explain reasoning for focusing only on phosphate fertilizers, and not referring to other types of fertilizers, such as nitrate fertilizers that could be applied.

A thorough review of the literature on strontium leaching into water resources via fertilizers was conducted during both the study and writing process. Another review of the literature found the same information: there are not many existing studies that are relevant to this study, and the ones that are relevant (which are already cited) only mention phosphate fertilizers.

Reviewer 4 Report

Thank you for the opportunity to revise this interesting manuscript.

I have found the study design appropriate, and the methods are clearly described.

Overall, the paper is interesting and very well presented, and the conclusions are supported by the findings.

For these reasons, I believe that the manuscript will be of interest to the readers, and it could be suitable for publication after a few minor revisions.

In particular, I have only two suggestions the authors can consider:

1) while reading the manuscript, even if it is very clear I have found excessive the use of acronyms within the text. I suggest the authors reduce the acronyms to the bare minimum, which would make the paper much smoother to read.

2) As I have understood, the population involved in the research has no information about the water quality they are supplied with (mostly private water sources). This issue is critical, but the authors have only mentioned this aspect in the conclusion section. I suggest deepening this key health issue, as the communication of drinking water quality has been clearly demonstrated to be fundamental in the population health risk perception. In particular, the lack of information could lead to a high population distrust, and the consequent use of alternative drinking water sources (e.g. streams, springs, and wells), which are not as monitored (and safe) as the public water supply. This aspect exposes people to health risks due to not safe water consumption, while this situation could be "easily" avoided with proper communication and information strategies. This reference could be helpful: Dettori, M.; et al. Population Distrust of Drinking Water Safety. Community Outrage Analysis, Prediction and Management. Int. J. Environ. Res. Public Health 2019, 16, 1004. It could be of interest if the authors could strengthen this aspect in the conclusion section, maybe considering some possible hypotheses of the communication strategies to implement in order to reach as much population as possible. 

I hope the authors will follow the suggestions, as I believe that the manuscript will be well received by the academic community.

Author Response

Reviewer 4:

Thank you for the opportunity to revise this interesting manuscript. I have found the study design appropriate, and the methods are clearly described. Overall, the paper is interesting and very well presented, and the conclusions are supported by the findings. For these reasons, I believe that the manuscript will be of interest to the readers, and it could be suitable for publication after a few minor revisions. In particular, I have only two suggestions the authors can consider:

  1. while reading the manuscript, even if it is very clear I have found excessive the use of acronyms within the text. I suggest the authors reduce the acronyms to the bare minimum, which would make the paper much smoother to read.

We have removed “(PDW)” from the title as well as all references to the “UN,” “CCL,” “NLCD,” and “USCB.” We also removed references to “HRL” and “VAHQWP” in several locations in the manuscript. As suggested, this has made the manuscript smoother to read.

  1. As I have understood, the population involved in the research has no information about the water quality they are supplied with (mostly private water sources). This issue is critical, but the authors have only mentioned this aspect in the conclusion section. I suggest deepening this key health issue, as the communication of drinking water quality has been clearly demonstrated to be fundamental in the population health risk perception. In particular, the lack of information could lead to a high population distrust, and the consequent use of alternative drinking water sources (e.g. streams, springs, and wells), which are not as monitored (and safe) as the public water supply. This aspect exposes people to health risks due to not safe water consumption, while this situation could be "easily" avoided with proper communication and information strategies. This reference could be helpful: Dettori, M.; et al. Population Distrust of Drinking Water Safety. Community Outrage Analysis, Prediction and Management. Int. J. Environ. Res. Public Health 2019, 16, 1004. It could be of interest if the authors could strengthen this aspect in the conclusion section, maybe considering some possible hypotheses of the communication strategies to implement in order to reach as much population as possible. 

I hope the authors will follow the suggestions, as I believe that the manuscript will be well received by the academic community.

Thank you for this detailed suggestion. First, the following sentence has been added to the first paragraph of 2.1 Data Collection to stress that households are mostly unaware of their PDW quality: “Survey data reveal that 80% of samples come from PDWs that have either never been tested or have been tested only once.” Second, the following sentence has been added to the Discussion section to address issues of risk, trust, and perceptions: “Other factors that may lead to underreporting social vulnerability are representative samples from populations who lack trust in the government, scientists, and institutions such as VAHWQP and those who perceive their PDW as ‘safe’ to begin with [46-47].” This includes references to the Dettori et al. (2019) article mentioned as well as a relevant article by Arcipowski et al. (2017). Finally, we have added the following to the Conclusion section to further highlight those important points: “In particular, consumers that should be targeted with clear information on the health risks of drinking unsafe water are those who cannot afford testing, renters, and those with perceptions that their water is ‘safe’ or who lack trust in science and institutions. This is critical as such populations may be unwittingly exposed to contaminants. Communication of science and health risks associated with drinking water is a field ripe for advancement—especially given recent international events such as Flint, Michigan. Thus, we argue for community-level approaches demonstrated by Dettori et al. [46], Arcipowski et al. [47], and Virginia Cooperative Extension’s VAHWQP in their county level PDW clinics and informational material (see https://www.wellwater.bse.vt.edu/vahwqp.php). These approaches of building a ‘trust ecology,’ combined with accessible resources for unbiased water quality testing (e.g., the tests conducted through VAHWQP), represent potential steps in the right direction.” These collective additions offer nuance, recommendations, and a way forward.